# A Flexible Tool to Correct Superimposed Mass Isotopologue Distributions in GC-APCI-MS Flux Experiments

**DOI:** 10.3390/metabo12050408

**Published:** 2022-04-29

**Authors:** Jennifer Langenhan, Carsten Jaeger, Katharina Baum, Mareike Simon, Jan Lisec

**Affiliations:** 1Bundesanstalt für Materialforschung und -Prüfung (BAM), Division 1 Analytical Chemistry, Richard-Willstätter-Straße 11, 12489 Berlin, Germany; jennifer.langenhan@bam.de (J.L.); carsten.jaeger@bam.de (C.J.); 2School of Analytical Sciences Adlershof (SALSA), Humboldt-Universität Zu Berlin, Albert-Einstein-Straße 5, 12489 Berlin, Germany; 3Max-Delbrück-Center for Molecular Medicine (MDC), Mathematical Modeling of Cellular Processes, Robert-Rössle-Straße 10, 13125 Berlin, Germany; katharina.baum@hpi.de (K.B.); mareike.simon@mdc-berlin.de (M.S.); 4Hasso Plattner Institute, Digital Engineering Faculty, University of Potsdam, Prof.-Dr.-Helmert-Straße 2-3, 14482 Potsdam, Germany

**Keywords:** mass isotopologue distribution, enrichment calculation, flux experiments, atmospheric pressure chemical ionization, R package, CorMID

## Abstract

The investigation of metabolic fluxes and metabolite distributions within cells by means of tracer molecules is a valuable tool to unravel the complexity of biological systems. Technological advances in mass spectrometry (MS) technology such as atmospheric pressure chemical ionization (APCI) coupled with high resolution (HR), not only allows for highly sensitive analyses but also broadens the usefulness of tracer-based experiments, as interesting signals can be annotated de novo when not yet present in a compound library. However, several effects in the APCI ion source, i.e., fragmentation and rearrangement, lead to superimposed mass isotopologue distributions (MID) within the mass spectra, which need to be corrected during data evaluation as they will impair enrichment calculation otherwise. Here, we present and evaluate a novel software tool to automatically perform such corrections. We discuss the different effects, explain the implemented algorithm, and show its application on several experimental datasets. This adjustable tool is available as an R package from CRAN.

## 1. Introduction

Mass spectrometry is a versatile analytical method that is widely used in clinical applications, metabolomics, chemistry, biochemistry, and elsewhere [1,2,3]. MS-based analyses are conducted using a variety of different methods and instruments with the commonality to separate and measure the mass-to-charge (m/z) ratio of molecules. MS is often used in combination with separation techniques such as gas chromatography (GC), liquid chromatography (LC), or ion mobility [1,4,5]. The separation of compounds in a sample by polarity, size, shape, or other characteristics facilitates the identification and quantification of different substances in mixtures and complex matrices [6]. The constant development improving individual aspects of MS, e.g., mass resolution, opens the door for novel applications.

One application particularly benefitting from higher resolution is metabolomics: the investigation of potentially all metabolites in a biological system to gain a better understanding of living organisms and to identify causes and potential cures for diseases. One of the aspects of metabolomics are metabolic flux experiments. In flux analysis, precursor molecules are labeled with isotopes that have a low natural abundance, provided to the biological system, and traced with respect to their conversion. Mostly, tracers containing ^13^C are used but ^15^N or ^18^O tracers are available as well. The techniques used to conduct such experiments include ^2^H/^13^C NMR and MS [7,8].

GC-MS is frequently applied in flux analyses, generally using electron impact ionization (EI) because large databases with EI spectra allow for a broader targeted evaluation. A more recent alternative is high-resolution GC-APCI-MS [9]. In comparison to EI, APCI is a soft ionization method leading to mainly [M+H]^+^ and [M]^+^ ion species, together with moderate fragmentations and rearrangements [9].

The so monitored incorporation of tracer molecules into the metabolome of biological samples is a common tool used to investigate metabolic pathways and processes. The incorporation of the tracer, e.g., ^13^C, is quantified by an analysis of the MID of a compound. Mass isotopologues (MI) are molecules with identical chemical structures but differences in mass due to the presence of different isotopes. The *rawMID* of a compound is a vector of measured ion intensities in a sample. The ratio of these ion intensities allows for the inference of the fractional amount of individual isotopes. In the case of ^13^C, the natural abundance corresponds to a fractional amount of about 1.1%. Accurate determination of the isotope labeling of the metabolites is a prerequisite for metabolic flux calculations. To this end, MIDs need to be corrected for natural abundance and various tools have been developed previously to achieve this correction including MIDcor, IsoCorrectoR, and IsoCor [10,11,12]. MIDcor corrects for the natural abundance of isotopes as well as for the potential overlap with other compounds, while IsoCorrectoR corrects for natural MID as well as for tracer purity (Appendix A).

In APCI experiments, we noticed (in addition to natural abundance and tracer impurity) additional effects requiring MID correction. Specifically, we observed reactions such as the loss of a proton, the loss of hydrogen, and the addition of water combined with the loss of CH_4_, as described previously [13]. The occurrence of such ionization products is molecule dependent and leads consequentially to superimposed MIDs, i.e., *rawMIDs* being an overlay of the same mass spectrum shifted by a small number of mass units. While the tools mentioned above allow researchers to correct MIDs for several effects, they do not provide a flexible framework to correct superimposed spectra resulting from different ionization adducts ([M]^+^, [M+H]^+^) and fragmentation reactions ([M+H_3_O−CH_4_]^+^).

Here, we describe such a function, CorMID, and demonstrate its usefulness by evaluating the spectra of a compound library to confirm the amount of fragment occurrence in APCI-MS and by applying it to biological samples of a flux experiment to improve data preparation for flux analysis.

## 2. Results and Discussion

Monitoring the incorporation of ^13^C-Glucose or similar tracer molecules into the metabolome of biological samples is a common tool used to investigate metabolic pathways and processes [8]. That includes unknown pathways, the speed and efficiency of metabolization, and the conversion of molecules normally not present in said biological systems. The ^13^C incorporation is quantified by an analysis of the MID of a compound. Mass isotopologues (MI) are molecules with identical chemical structures but differences in mass due to the presence of different isotopes, here either ^12^C or ^13^C. The mass difference between both carbon isotopes is 1.003355 Dalton. Neglecting the contribution of other chemical elements in a compound, the number of carbon atoms defines the number of possible MI. Glucose, which is frequently used as a tracer molecule, contains six carbon atoms, leading to seven possible MI ranging from 180.0634 Da (isotopologue M0, containing zero ^13^C and six ^12^C atoms) to 186.0835 Da (M6, containing ^13^C = 6 and ^12^C = 0). Often, fully labeled Glucose ([U-^13^C]Glc) is provided to cells for uptake and its carbon atoms are distributed to glucose-derived metabolites via the metabolic pathways. The amount of labelling in a compound observed during the experiment is indicative of the amount produced from the tracer molecule, which ultimately facilitates the reconstruction of a metabolic network and the ability to infer knowledge about cellular processes.

A key step during the necessary data processing is the correction for naturally abundant ^13^C, which is present in the environment at about 1.1%. As naturally abundant ^13^C is randomly distributed in compounds, we measure for an analyte containing three carbon atoms, e.g., lactic acid, about 3.2% and 0.04% of the intensity of the M+0 peak at M+1 and M+2, respectively (the measured intensity of M+3 is negligible). M+0 is the ion intensity of the monoisotopic peak of an analyte. M+i are the ion intensities of the isotopologue peaks of the molecule with approximately i×1 Dalton mass difference to M+0.

CorMID uses a fragment distribution vector (*r*) and the corrected MID (*corMID*) to determine a reconstructed vector of similar size (*recMID*). By finding the *recMID* that is closest to the observed vector of measured ion intensities (*rawMID*), the correct *corMID* is determined (Appendix A). Correcting a *rawMID* for natural abundance means subtracting the stochastically expected intensity of M+1 and M+2 before normalizing to the summed intensity, which would lead to a *corMID* of {1, 0, 0} in this case.

While this correction seems trivial, several effects complicate the calculations in real experiments: derivatization, fragment and adduct formation, and combinations of the latter two. Derivatization is the process where, e.g., tri-methyl-silyl groups (TMS) are attached to molecules of biological origin during sample preparation to facilitate volatilization. Each TMS group contains 3 C with a stochastic labelling of 1.1%. Furthermore, it contains 1 Si, which is either present as ^28^Si (92.2%), ^29^Si (4.7%), or ^30^Si (3.1%). While isotopes of other atomic species can mostly be neglected due to their low abundance (e.g., ^15^N~0.004%), TMS molecules contribute strongly to a compound’s MID and must be considered during calculations (Figure 1B). Incorporation of ^13^C from a tracer molecule will affect only the carbon of biological origin and not the carbon contained in TMS groups (Figure 1C). Fragmentation describes the process where APCI results in the formation of different in-source fragments. Generally, the protonated version [M+H]^+^ is observed. However, [M+H]^+^ ions are often prone to proton losses, leading to [M]^+^ and [M-H]^+^ ions. The fractional distribution of these ion entities is dependent on the molecule and instrument conditions, but it is stable within measurement series (own observation). A fourth fragment that is often present in spectra of carbonic acids is formed by attaching water and by dissociating a methane molecule [M+H_3_O−CH_4_]^+^. The resulting ion is in close proximity to the M+2 isotopologue (H_2_O ~18 Da, CH_4_ ~16 Da, difference to M+2 ~−30 mDa).

Several software packages exist to correct for naturally abundant ^13^C, mostly assuming the presence of either a single or two ion entities ([M+H]^+^ and [M]^+^) [10,11]. However, as described above, many molecules give rise to several fragment ions which overlap with respect to their MIDs. The measured intensity at [M+H]^+^ is the sum of [M+H]^+^, [M]^+^+1 and [M-H]^+^+2. This is due to the mass resolution (R = m/Δm) of current quadrupole time-of-flight instruments being around 35,000. CorMID is designed to mitigate that disadvantage. Devices with a higher resolution, e.g., Fourier-transform ion cyclotron resonance instruments, (R ~1 × 10^6^) will resolve the isotopic fine structure of the compound and allow for the separation of these peaks; here, CorMID is not used. Neglecting the contribution of multiple fragments will lead to severe errors in the correction of natural abundance and calculation of tracer incorporation. As an example, Glucose (Glc) forms approximately 10% [M-H]^+^ within the APCI spectrum. While this is not critical in standard metabolomics experiments, where the intensity of [M+H]^+^ would be compared over samples, this will lead to a 10% [M+H]^+^+4 peak in an experiment where fully labeled Glc is used as a tracer. Not accounting for the [M-H]^+^ fragment in consequence will lead to a *corMID* of {0, 0, 0, 0.1, 0, 0.9) or M4 = 10% and M6 = 90%. We have programmed a function CorMID that estimates the ratio (fractional distribution) of potential fragments and the corresponding corrected MID, solely based on the measured *rawMID* and the sum formula of the compound. In the following paragraphs, we show the problem, and explain and test the algorithm on a large set of standard compounds as well as on a biological data set from a cell culture experiment using fully labeled Glc and glutamine (Gln) as tracer molecules.

### 2.1. Library Approach

We have measured a comprehensive compound library on a Bruker HR qTOF MS coupled to an Agilent 7890B via APCI. Of the 604 compounds, 367 compounds were detectable using GC-APCI. All these compounds contain ^13^C only at the level of 1.1% (natural abundance), i.e., correcting for natural abundance should lead to an estimation of M0 as 100% in every case. If we apply the newly developed CorMID function to all 367 compounds, we obtain M0 ≥ 99% in 310 of all cases (84%) and M0 ≥ 95% in 333 cases (91%) (Figure 2). The 34 compounds (9%) where M0 < 95% show widely spread values (Figure 2). Manual inspection of their spectra revealed that, in almost all cases, spectral impurities were present leading to wrong spectra (Appendix A).

We can assume that the estimated fragment distribution *r* for the 288 compounds, which were determined to be fully non-labeled (M0 = 100%), is correct. Therefore, we analyzed the occurrence of fragments in these compounds (Figure 3). In total, 159 compounds (55%) show fragments other than [M+H]^+^ in significant amounts (>5%). This is generally not a problem for data analysis in experiments without labelling. However, in flux experiments, it will hamper data evaluation. We can estimate the error in the MID corrections we encounter (when not considering fragments other than [M+H]^+^) by providing a fixed fragment ratio as a parameter to the function, i.e., we pretend that [M+H]^+^ = 100% and calculate *corMID* under this assumption. We find a strong deviation (M0 ≤ 95%) in about 47% (*n* = 136) of all cases. All 33 compounds that showed the fragment [M+H_3_O−CH_4_]^+^ yielded a wrong M0 when the fragment was not considered. The same was true for most of the compounds showing [M]^+^ and [M-H]^+^ (Appendix A). When performing MID corrections in tracer experiments without considering relevant fragments, a 5% occurrence of a [M-H]^+^ fragment would lead to a spurious 5% estimate of the M(n−2) isotopologue of a compound labeled with n carbon atoms and, in consequence, lead to wrong flux estimates.

### 2.2. Application to Biological Data

To test this hypothesis and the performance of the correction function, we applied it to a targeted metabolite list from a biological data set. Here, cancer cell cultures were labeled with either [U-^13^C]Glc or [U-^13^C]Gln for 24 h in five replicates each before MS analysis. We demonstrate the general functionality of the function using measured glutamine intensity as an example (Figure 4).

We can assume that Gln is non-labeled in the Glc-tracer experiment and fully labeled in the Gln-tracer experiment. We can further estimate from the *rawMID* (Figure 4) the approximate distribution of potential fragments ([M-H]^+^~10%, [M]^+^~45, [M+H]^+^~45%) but have no chance to calculate it directly as we do not know if M0 is truly 100% (i.e., if Gln is non-labeled via [U-^13^C]Glc in this experiment). Applying our function CorMID to estimate the fragment distribution *r* and *corMID*, we obtain the values presented in Table 1.

Clearly, Gln is determined to be non-labeled in the five samples from the Glc-tracer experiment and approximately 90% labeled in the five samples from the Gln-tracer experiment. The occurrence of approximately 10% of M4 could be an impurity of the tracer but is of biological origin here, as we checked tracer purity independently. Moreover, sample S10_Gln is nearly fully labelled, which is confirmed by looking at the difference in *rawMID* (Appendix A). The approach generally gives reproducible results over replicates. The remaining differences in parameter estimation are likely due to biological variance over replicates.

However, when we extended this analysis to 36 metabolites from this data set, we occasionally observe the parameter estimation to yield alternative solutions for individual samples, similar to S10 in Figure 5 (Appendix A). This is the case for 7 out of 36 metabolites. One further example is depicted in Figure 6, where 2-oxo glutaric acid is consistently M5 and M3 labeled in Gln-samples, and M1 labeled in Glc-samples when the fragment distribution *r* is fixed in calculations (Figure 6B). When *r* is estimated together with the MID (Figure 6A), the algorithm provides different solutions for samples S3 and S4. The reason for this discrepancy is the presence of comparable minima in the solution space, both explaining the measured intensities well. Here, the presence of high intensity at the M+0 position is attributed to the occurrence of an M2 labeled [M-H]^+^ fragment.

Due to the underdetermined equation system, such ambiguities are hard to overcome algorithmically (Appendix A). A practical and robust solution is to estimate fragment occurrences and corrected MIDs independently. Our function allows us to provide both a fixed fragment distribution vector as well as a fixed MID, in which case the data are fitted only with respect to the unknown component (Appendix A). Using compound spectra obtained from samples without artificial labelling, one can fix the MID at M0 = 100% to estimate the best fitting fragment distribution. This fragment distribution in turn can be used to estimate corrected MIDs in labeled samples under the assumption that fragment formation for a specific compound and the MS system are stable and independent of the present isotope.

## 3. Materials and Methods

The algorithm behind CorMID is a greedy minimization of a residual error (*err*) between an observed vector of measured ion intensities (*rawMID*) and a reconstructed vector of similar size (*recMID*) calculated out of a fragment distribution vector (*r*) and the corrected MID (*corMID*) (Appendix A). The size of *r* is generally dependent on the number of fragments considered, with currently *k* = 4 possible fragments defined and $\overset{4}{\underset{k=1}{\sum }}r_{k}=1$. The size of vector *corMID* is dependent on *n*, the number of biological carbon atoms within the molecule. It is of size *n* + 1, with $\overset{n}{\underset{i=0}{\sum }}M_{i}=1$ and $M_{i}\in \left[0,1\right]$. When *M*_0_ = 1, the compound is fully unlabeled and fully labelled when *M*_n_ = 1. For every *r* and *corMID*, we can reconstruct a vector *recMID* of similar size as *rawMID* to compute $err=\sqrt{\sum {\left(rawMID-recMID\right)}^{2}}$. The reconstruction is performed by converting *corMID* into a matrix and by multiplying this matrix with *r*, *recMID* = *f*_rec_(*corMID*, *n*, *r*)·*r*. The matrix dimensions are dependent on the measured intensities in *rawMID*, the number of biological carbon atoms *n*, and the specified fragments in *r*. To this end, the function *f*_rec_() will prepend and append *corMID* with 0 values depending on the present fragments. The following multiplication with *r* will yield a vector of size *rawMID*. For example, if *r* contains only two fragments [M]^+^ and [M+H]^+^, *n* = 2 and *rawMID* contains four consecutive ion intensities starting at [M]^+^, the result of *f*_rec_() would be the matrix {*corMID*,0}{0,*corMID*}. Because the size of *corMID* is *n* + 1, multiplying this matrix with *r* will lead to *recMID* being of a size similar to *rawMID*. In fact, *recMID* and *rawMID* are a superimposed *corMID* after being shifted according to *r*.

The minimization of *err* has a unique solution when either *r* or *corMID* are fixed. To provide an example, let us assume that *r* is known and stable for a specific compound with two carbon atoms of biological origin. The optimal *corMID* fitting to this *r* and a specific *rawMID* can then be obtained by testing all possible combinations of *corMID*, i.e., all vectors *M*. In practice, the solution is approached by testing possible *M* using a widely spread set of potential solutions initially (*M* = {1,0,0}, {0,1,0}, {0,0,1}, {0.5,0.5,0}, {0.5,0,0.5} and {0,0.5,0.5}) to find the best fit (minimal *err*). In this example, a step size of 0.5 was used to build the set of solutions. In each following iteration, a new set of potential solutions is tested based on the best fit of the previous round and decreasing the step size.

If both *r* and *corMID* need to be estimated in parallel, the above approach is performed in a nested fashion, i.e., testing each *corMID* hypothesis *M* with a solution set of different *r*. This can lead to wrong *corMID* estimates as, e.g., the constructed *recMID* vector in the above setting (*n* = 2, two fragments for r) with *corMID =* {*0*,*1*,*0*} and *r =* {*1*,*0*} is exactly the same as for *corMID =* {*1*,*0*,*0*} and *r =* {*0*,*1*}. When *r* can neither be estimated from non-labeled control samples nor be estimated based on statistical evaluation of replicate measurements, the function CorMID offers the option to set a penalization parameter, which will apply a weighting factor on *err* based on the amount of [M+H] estimated for *r*. Penalizing solutions with small amounts of [M+H] is justified based on Figure 3. For further details, please see the package documentation.

To test CorMID systematically, the MSMLS metabolite library containing 604 unique compounds was purchased from Sigma-Aldrich (Hamburg, Germany). Metabolite standards (5 µg) were reconstituted by adding 100 µL 5% methanol or 100 µL 3:3:1 (*v*/*v*/*v*) chloroform/methanol/water, respectively, to each well of 96-well plates according to the manufacturer’s instructions; 15–24 metabolites were combined each into *n* = 28 different master mixes, aliquoted, and dried down in an Alpha 2–4 vacuum rotator (Christ, Osterode, Germany). All samples were derivatized to substitute reactive protons with tri-methyl-silyl groups and measured on an Impact II Q-TOF MS (Bruker, Bremen, Germany) coupled with an Agilent 7890 B gas chromatograph (Agilent, Waldbronn, Germany) via a GC-APCI II source (Bruker, Bremen, Germany). Data processing was carried out as previously described [14].

To test CorMID on complex biological samples, we used previously published data from a flux experiment in colon cancer cell cultures applying glucose and glutamine as tracer molecules to incorporate labelled carbon. Here, SW620 and SW480 cells (ATCC, Manassas, VA, USA) were grown, propagated, treated, harvested, and prepared as previously described [15].

## 4. Conclusions

We showed that in many biologically relevant standard compounds (159/288 or 55%), fragments other than the predominant [M+H]^+^ are present in significant amounts (>5%) which lead to errors during MID correction if unaccounted for. We introduced the R package CorMID, a tool to disentangle the superimposed *rawMID* into the likely fragment distribution *r* and the correct MID in parallel. We demonstrated the functionality of CorMID on spectra from a large compound library and biological samples of two independent tracer experiments. It is also possible to adjust CorMID to take fragments other than [M+H]^+^, [M]^+^, [M-H]^+^, and [M+H_3_O−CH_4_]^+^ into account depending on the used method and molecules. Therefore, the introduced function is also of interest for other ionization methods such as electrospray ionization, which creates adducts and fragments interfering with MID corrections.

## Figures and Tables

**Figure 1 metabolites-12-00408-f001:**
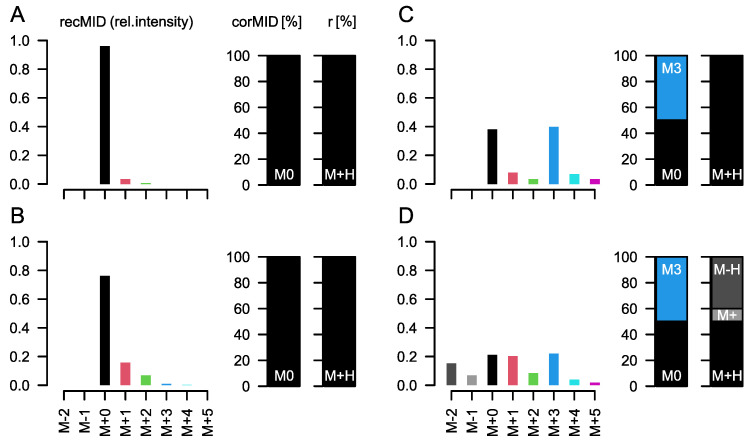
In situ demonstration of the problem of a superimposed MID. Each subpanel shows the relative measured intensities (*recMID*, **left**) of a compound reconstructed assuming a specific labelling state (*corMID*, center) and a specific ion formation (fragment ratios *r*, **right**). (**A**) For a compound with *n* = 3 carbon atoms, e.g., lactic acid, forming a single molecular ion ([M+H]^+^) and showing only natural abundance of ^13^C we would measure ion intensities at M+0 (which is the [M+H]^+^ peak here) and at M+1 with approx. 3% intensity of M+0. Here, M represents the atomic mass of the molecular ion of the compound and the amended number indicates the total number of ^13^C atoms contained in the specific MI. No ^13^C from a tracer molecule is incorporated (M0 = 100%). In GC-APCI-MS, metabolites are usually silylated before analysis. The Si and C atoms incorporated during the silylation affect the *recMID* (**B**). If 50% of the lactic acid is fully (M3) labelled during a tracer experiment, *recMID* and *corMID* would be similar to the values presented in (**C**). (**D**) When effects such as proton loss (here, 10% of [M]^+^ and 40% of [M-H]^+^) are present, the *recMID* becomes more complex even for a simple *corMID* consistent with (**C**).

**Figure 2 metabolites-12-00408-f002:**
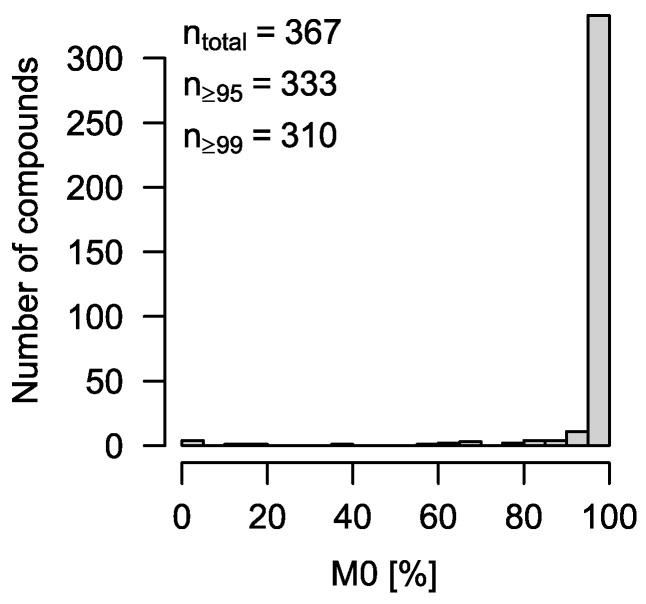
Histogram of estimated M0 for 367 library spectra. Ideally, all compounds are M0 = 100%. However, due to spectral impurities, this is only true for 288 compounds, or 78%. Still, more than 90% of all compounds (*n* = 333) show an automatic solution with M0 ≥ 95%.

**Figure 3 metabolites-12-00408-f003:**
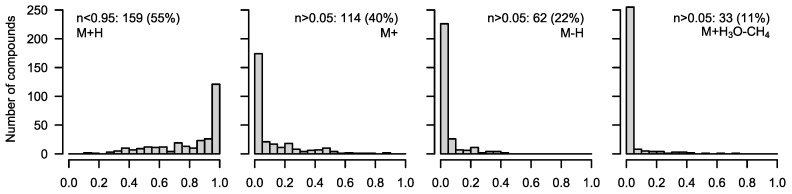
Occurrence of different fragments in 288 standard compounds of the library, which were estimated to have 100% M0 (only naturally abundant ^13^C). A nearly exclusive formation of [M+H]^+^ was found in 129 compounds, while in 159 compounds, other fragments made up >5% of the total frequency. Proton losses in significant amounts occurred more often ([M]^+^ = 114, [M-H]^+^ = 62) than the formation of [M+H_3_O−CH_4_]^+^ (33).

**Figure 4 metabolites-12-00408-f004:**
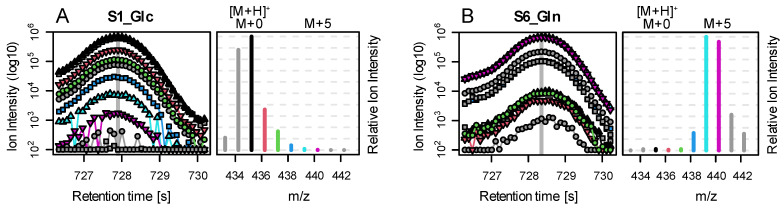
Base peak chromatogram (BPC, **left**) and *rawMID* (**right**) for two example files from the biological data set. For both samples, we show the Gln peak: (**A**) after 24 h of Glc labelling and (**B**) after 24 h of Gln labelling. The *rawMIDs* are extracted from the respective BPC scan indicated by a vertical grey line. BPCs, and their corresponding *rawMID* relative intensities are color coded as follows: black indicates [M+H]^+^; red, green, blue, light blue, and purple indicate [M+H]^+^+1 to 5; and grey indicates [M-H]^+^, [M]^+^, and [M+H]+6/7. Absolute intensities in BPCs are depicted in log10-scale, and *rawMIDs* are normalized relative to the highest ion peak in the MID. In sample S1 (Glc as tracer and Gln putatively non-labeled), we observe that the [M+H]^+^ ion (black) shows the highest ion abundance. However, the *rawMID* represents an overlay of the MIDs of three adducts: [M-H]^+^, starting at m/z~433; [M]^+^, starting at m/z~434 (both grey); and [M+H]^+^, starting at m/z~435 (black). In consequence, the peak intensity at m/z~435 is actually the sum of three isotopes: [M-H]^+^+2, [M]^+^+1, and [M+H]^+^+0. In sample S6 (Gln as tracer and putatively fully labeled), we observe a clear shift in the MID by 5 mass units towards the right (caused by five ^13^C labeled carbon atoms in Gln). However, to estimate the correct amount of labelling, we need to consider tracer impurity, natural abundance of Gln, as well as the C and Si atoms introduced by silylation and three relevant fragments (adducts).

**Figure 5 metabolites-12-00408-f005:**
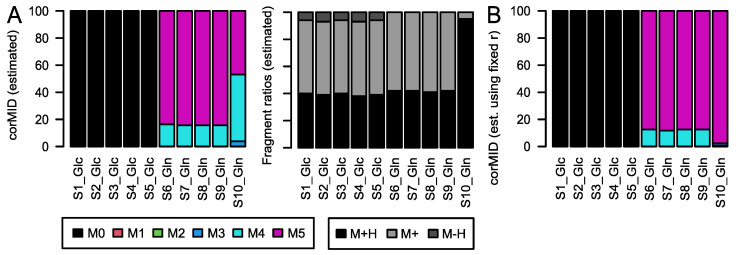
*corMID* and fragment distribution *r* for Gln after Glc labeling (S1–5) and after Gln labeling (S6–10). (**A**) *corMID* and *r* estimated from the measured intensities. (**B**) *corMID* estimated using the median values from (**A**) as a fixed *r*. The according *rawMIDs* for all files can be found in Appendix A.

**Figure 6 metabolites-12-00408-f006:**
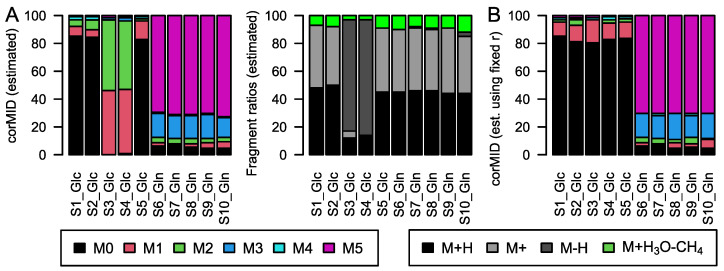
(**A**) *corMID* and fragment distribution for 2-oxo glutaric acid after Glc labeling (S1–5) and after Gln labeling (S6–10). (**B**) *corMID* intensities of 2-oxo glutaric acid using a fixed fragment distribution based on the values obtained in (**A**). Note that samples S3 and S4 show different solutions when *r* and *corMID* are estimated in parallel while estimating *corMID* alone (*r* fixed) leads to robust results. The according *rawMIDs* for all files can be found in Appendix A.

**Table 1 metabolites-12-00408-t001:** Estimated *corMID* for Gln in 10 biological samples, either labeled for 24 h with [U-^13^C]Glc or supplied for 24 h with [U-^13^C]Gln (see column header). The average estimated fragment distribution used was [M+H]^+^ = 38.1%, [M]^+^ = 56.4%, [M-H]^+^ = 5.5%, and [M+H_3_O−CH_4_]^+^ = 0%.

	S1_Glc	S2_Glc	S3_Glc	S4_Glc	S5_Glc	S6_Gln	S7_Gln	S8_Gln	S9_Gln	S10_Gln
M0	100	100	100	100	100	0	0	0	0	0.78
M1	0	0	0	0	0	0	0	0	0	0
M2	0	0	0	0	0	0	0	0	0	0
M3	0	0	0	0	0	0	0	0	0	1.56
M4	0	0	0	0	0	12.50	11.72	12.50	12.50	0
M5	0	0	0	0	0	87.50	88.28	87.50	87.50	97.66

## Data Availability

The *rawMID* data from the biological data set used in Section 2.2 of this manuscript were provided in the R package CorMID.

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
