# Peer review of "A Flexible Tool to Correct Superimposed Mass Isotopologue Distributions in GC-APCI-MS Flux Experiments"

_metabolites, 2022, doi:10.3390/metabo12050408_

Round 1

Reviewer 1 Report

In this manuscript the authors carefully explain and characterize the problematic behind performing flux analysis using GC-APCI-MS, they also present a smart computational tool to overcome such problem.

In general the text is well-written, but I found it hard to follow in some paragraphs and was forced to guess the what the authors really wanted to convey. Given the quality of the computational tool, I believe the text could present it better with some minor improvements (see minor issues).

In addition I would like the authors to assess the issues below.

> Major issues:
- In line 61, the authors state that GC-APCI-MS "is beneficial for the annotation of unknown compounds". I strongly disagree. The metabolomics challenge of unknown identification is the same for LC-MS and GC-CI-MS, even harder if derivatization is taken into consideration and adding the fact that MSMS woud be needed. In addition, identification of unknown compounds is not the aim of the tool presented. This statement should either be extended or, at least, referenced.

- Regarding the discussion in lines 150-151, there is no reference made to data from instruments with mid-top resolution levels, eg. Orbitrap 60/120/240K R @200m/z. Resolution, as it is indicated in the text, indeed has a strong impact on how well the ion peaks are resolved and thus affects the  peak finding and quantification step. Though, derivatization with TMS and MEOX increases significantly the m/z of some compounds (eg. sugars) and then the m/z fall within the lower resolution capability, difficulting the separation of Si and C isotopologues. In other words, for some compounds, the input of CorMID should consider whether the isopologues are more or less well-resolved according to the mass-analyzer conditions. Does CorMID consider such problematic?

> Minor issues:

- There is some lack of detail in the introduction of MID concept in line 63, in which the authors jump into the "mathematical" concept while not commenting on the biochemical origin or chemical result (isotopologues) of the so-called "vector".  Later, they introduce a very good example at the start of section "2. Results and Discussion".
I strongly believe the mentioned paragraph of the introduction could be written for a better understanding, especially for readers who are not experts in labelling or flux analysis experiments.

- Lack of Figure S1 caption in the supporting information docx document.

-  Some other minor inconsistencies and imprecisions are found in the text. For example, in the case of line 92 and line 109, I wonder why the mass difference between C12 and C13 is stated in two different manners.
- In lines 107-109, one could use the specific term "monoisotopic" to refer to the non-labelled M+0 is the ion/peak and M+i are the isotopologues.
- In line 111, authors refer to "this rawMID" while this acronym was not previously defined, should it be "recMID"? These and other acronyms are later described int the methods section, but the context in line 111 is missing. 
- In line 130, authors should also consider other silyl-derivatization methods such as TBDMS. 

>Text writing issues:
- Line 40, "were" should be "where".
- Line 50, "flux analyses" should be "flux analysis"
 - Consider revising the use of excessive connectors and commas as these difficult conveying the message of the text. For example lines (180-185). 

Reviewer 2 Report

Manuscript by Langenhan et al. describes the development and implementation of a new software tool for the correction of mass isotopologue distributions (MID). It is well known that MIDs must be corrected for natural abundance prior to analysis of enrichments and mathematical modeling of metabolic flux. However, what is less appreciated, is the fact, that different ionization techniques may introduce another layer of complexity, such as adducts, which require additional corrections. Inability to account for these overlapping MIDs would result in erroneous metabolic flux estimates, because each isotopologue can be composed of different mass shifts depending on the adduct it originates from. Therefore, the topic is of interest to the metabolic community. Currently, a number of software tools for MID correction exist, and they offer a different degree of functionality regarding correction for natural abundance or adducts. However, the comprehensive nature of ‘CorMID’ goes a step further by incorporating variable adducts, and could become a useful tool for metabolic researchers. While the paper is technically sound, the presentation of the problem, the algorithm, and tool features could be substantially improved. I have few comments which should be addressed prior to considering this manuscript for publication.

Major comments:

  1. The name of this package ‘CorMID’ is very similar to ‘MIDcor’ (PMID: 28158972). Did Authors considered a different name? While it is not a technical issue, authors should consider a potential problem of confusing users by similar package names. Perhaps it would be advantageous to have a more specific name for this dedicated software tool. What authors think about “CleanMID” ? Would such name capture the essence of this package and its ability to correct for both natural abundance and superimposed MIDs?
  2. The manuscript describes the process of correction, but it does so in a somewhat convoluted way. For example, it was hard to remember the relationship between rawMID, recMID, corMID while reading the paper. Also, it seems that recMID and rawMID are used interchangeably sometimes e.g. page 3, lines 110-111. Clarification of the definitions and consistency in naming is necessary.
  3. I highly recommend that authors put more emphasis on efficient commutation of the procedure itself. This manuscript would greatly benefited by incorporation of a schematic which would guide reader (or user) though individual steps of the algorithm. For example, how the process is affected depending on how r is estimated (issue related to the process described in Page 7, lines 249-268). Authors could create schematic similar to the algorithm scheme in Figure 5 of this publication (PMID: 28208288) which would summarize how the tool exactly works.
  4. Page 2, lines 53-56. While understandable that the authors focus here on the MS based approaches since their software package is applicable to MS data, it should be mentioned that many flux experiments are conducted using NMR spectroscopy which is position and isotope specific and thus allow to resolve simultaneous tracers experiments. Here are some example papers on 15N NMR (PMID: 28208288) and 2H/13C NMR (PMID: 31891762) and flux estimations.
  5. Page 2, lines 67-72. Authors could expand on this fragment of the text. A table comparing newly developed software package to the other similar existing tools would be a good addition. Which features have overlap and what is new in current tool would be a nice highlight of authors work.
  6. Page 1, lines 38-49. Is this paragraph necessary? After all, the software tool is only dedicated to tracer experiments, not untargeted metabolomics. I understand that authors cite their previous work on APCI. However, in the current form, neither this part of the text nor references seems particularly relevant to the paper and authors never use them in discussion. I suggest removing this paragraph and focusing on the tracer related content.

Minor comments:

  1. Page 2, lines 59-62. Minimal fragmentation may be advantageous for annotation of unknown compounds, but keep in mind that generation of multiple fragment for a metabolite is advantageous for flux analysis, (PMID: 22510303)
  2. Page 2, lines 65-67. This sentence is too simplistic. Yes, knowledge of enrichments is required for flux calculation but may not be sufficient. Please modify this sentence. Consider e.g. “Accurate determination of the isotope labeling of the metabolites is a prerequisite for metabolic flux calculations. “
  3. Page 2, line 77. Consider explaining better what is superimposed here. At the first read this part was slightly confusing for me.
  4. Page 2, line 98. Please change the tracer naming to match the standard convention, i.e. [U-13C]Glc etc.
  5. Page 6, line 212 and Figure 4. The grey line is very light and easy to miss. Consider using a different color or contrast which will make it easier to see.

Reviewer 3 Report

This is a very interesting submission regarding a flexible tool (R-package) to correct superimposed MS isotopologue distributions in GC-APCI-MS. With this submission, authors continue their contribution on the field. The paper is well written. The experimental part is presented well and there is no major scientific objection or question. Supplementary material is very useful. my only suggestion has to do with the need for some additional references and relevant discussion in Introduction and/or Results, mainly referring to similar approaches, even in LC-MS.
